# Coherent superpositions of three states for phosphorous donors in silicon prepared using THz radiation

S. Chick[1], N. Stavrias[2], K. Saeedi[2], B. Redlich[2], P.T. Greenland[3], G. Matmon[3], M. Naftaly[4], C.R. Pidgeon[5], G. Aeppli[6,7,8] & B.N. Murdin[1]

Superposition of orbital eigenstates is crucial to quantum technology utilizing atoms, such as atomic clocks and quantum computers, and control over the interaction between atoms and their neighbours is an essential ingredient for both gating and readout. The simplest coherent wavefunction control uses a two-eigenstate admixture, but more control over the spatial distribution of the wavefunction can be obtained by increasing the number of states in the wavepacket. Here we demonstrate THz laser pulse control of Si:P orbitals using multiple orbital state admixtures, observing beat patterns produced by Zeeman splitting. The beats are an observable signature of the ability to control the path of the electron, which implies we can now control the strength and duration of the interaction of the atom with different neighbours. This could simplify surface code networks which require spatially controlled interaction between atoms, and we propose an architecture that might take advantage of this.

[1] Advanced Technology Institute and SEPNet, University of Surrey, Guildford GU2 7XH, UK. [2] Radboud University, Institute for Molecules and Materials, FELIX Laboratory, Toernooiveld 7c, 6525 ED Nijmegen, The Netherlands. [3] London Centre for Nanotechnology and Department of Physics and Astronomy, University College London, London WC1H 0AH, UK. [4] National Physical Laboratory, TQEM, Hampton Road, Teddington, Middlesex TW11 0LW, UK. [5] Institute of Photonics and Quantum Science, SUPA, Heriot Watt University, Edinburgh EH14 4AS, UK. [6] Laboratory for Solid State Physics, ETH Zurich, Zurich CH-8093, Switzerland. [7] Institut de Physique, EPF Lausanne, Lausanne CH-1015, Switzerland. [8] Swiss Light Source, Paul Scherrer Institut, Villigen PSI CH-5232, Switzerland. Correspondence and requests for materials should be addressed to S.C. (email: s.chick@surrey.ac.uk) or to B.N.M. (email: b.murdin@surrey.ac.uk).

In the silicon donor impurity phosphorus, the outermost electron is bound to the ion at low temperature with hydrogen-like states and a level spectrum according to the Kohn–Luttinger (KL) model[1]. The energy and length scales are determined by the dielectric constant and the effective mass, so the transitions occur in the region of 10 THz and the effective Bohr radius is $a_0^* = 3$ nm. There is a long history of incoherent spectroscopy of shallow impurities (P, Bi and so on) in silicon, normally utilizing the Fourier Transform Infrared technique (FTIR) (Si:Bi[2]; Si:P[3]; and more recently our own high field (30 T) work[4] in Si:P). Si:P can now be positioned and detected with atomic precision[5] to make devices that demonstrate control over coherent quantum interactions between neighbouring donors[6,7] and semiclassical interaction with contacts[8]. Optical control of Rydberg orbits of trapped atoms in vacuum has provided high-fidelity gate interactions[9] and fine control over the electron trajectory[10], with applications in atomic clocks and quantum computing[11,12]. Similar experiments have been carried out on optical transitions in semiconductors[13], but it is known that in free space atoms one is able to gain better control by including many orbital eigenstates in the wavepacket[10,13]. The importance of control over neighbouring qubit interactions in silicon has been shown separately[6,7,9,13], and it is also appreciated that control over interactions is important for the implementation of surface code error quantum correction[14–17]. A convenient radiation source for coherent studies in the THz spectral region is the free-electron laser. Particularly relevant in this regard are Rabi oscillations reported on the $1s \to 2p_+$ system of hydrogenic donors in GaAs in a magnetic field[18] and the observation of the excitonic Autler–Townes effect, utilizing InGaAs/AlGaAs quantum wells[19].

The present work bridges these previous studies by investigating three-eigenstate control of phosphorous orbitals in silicon. An interferometric experiment is implemented using a free-electron laser as the source, targeting the $1s(A_1) - 2p_\pm$ orbital transitions under a small magnetic field. A three-eigenstate superposition is generated, and its time evolution demonstrated through direct observation of beats in the interferogram. Measurements in the Si:P system are compared with the theoretical results for both the simple hydrogenic sytem and the multi-valley effective mass approximation appropriate for the crystal. We find that while the multi-valley effective mass theory is complicated, given simplifications appropriate for our experimental conditions the results are directly comparable to the simple and intuitive hydrogenic picture.

## Results

**Hydrogen three-level superpositions.** Orbital quantum control of the hydrogenic states of isolated Si:P donors with THz pulses has so far been restricted to simple two-level superpositions[20,21] as in the example of $1s$ and $2p_0$ states. We illustrate this with the simpler hydrogen $1s$ and $2p_0$, and their superpositions shown in Fig. 1a,b; see next section for donor wavefunctions in Si. In a superposed state, the relative phase of the two contributing hydrogen eigenstates evolves as a function of time resulting in oscillation of the centre of mass. Oscillations of different symmetry may be obtained, for example, with superposition of $1s$ and $2p_-$ states by a circularly polarized pulse, shown in Fig. 1c. Building wavepackets out of multiple states provides a greater variety of possible overlaps and spatio-temporal evolution. For example the $1s$, $2p_+$ and $2p_-$ states form a three-eigenstate system that allows freedom of control over the probability density in the x–y plane. To produce an arbitrary superposition state with a single pulse requires complete control of the polarisation and pulse shape. The control can be simplified significantly if the

degeneracy of the $2p_\pm$ states is lifted with a small applied magnetic field, in which case a pair of linearly polarised pulses with variable delay is sufficient for most purposes. The superposition amplitudes, $|\psi\rangle = \psi_0 |1s\rangle + \psi_- |2p_-\rangle + \psi_+ |2p_+\rangle$, may be found analytically in the case of long dephasing times. Consider a Zeeman splitting of the excited state $2\hbar\Delta$ about a central frequency, $\omega$, and a coherent, resonant, x-polarized pulse with Rabi frequency $\Omega$. If the splitting is small ($\Delta \ll \Omega, 1/t_p$, where $t_p$ is the pulse length) then at time $t_d$ in the dark after the pulse

$$\psi_0 = c \text{ and } \psi_\pm = -is e^{-i(\omega t_d \mp \Delta t_d)}/\sqrt{2} \quad (1)$$

where, $c = \cos(\Omega t_p/\sqrt{2})$, $s = \sin(\Omega t_p/\sqrt{2})$. We applied the rotating wave approximation and ignored a global phase and all dephasing/relaxation. Figure 1d shows the evolution of $|\psi\rangle$ in the dark for the case where $s = 1/\sqrt{2}$. The fast oscillation at frequency $\omega$ is shown across each row on Fig. 1d, and the slower evolution at frequency $\Delta$, from row to row. As is clear from Fig. 1d, the transverse component of the dipole moment of the superposition is $\langle \mu_y \rangle = \mu_0 sc \sin \omega t_d \sin \Delta t_d$, where $\mu_0$ is a constant, so the intensity of the transverse luminescence at frequency $\omega$ oscillates slowly in the manner known as quantum beating[22], often used for extraction of $\Delta$ through its Fourier transform (FT)[23]. Greater flexibility in state preparation is provided by a second pulse. If it is identical to the first and arrives at time $t_d$, it produces

$$\psi_0 = 1 - s^2\left(1 + Ce^{-i\omega t_d}\right), \psi_\pm = s\left[\pm S - ic\left(C + e^{i\omega t_d}\right)\right]e^{-i(\omega t_d + \omega t \mp \Delta t)}/\sqrt{2} \quad (2)$$

where $C = \cos(\Delta t_d)$, $S = \sin(\Delta t_d)$, and $t$ is the time in the dark after the second pulse. In this case the quantum beat may be observed in the absorption, which is proportional to

$$1 - |\psi_0|^2 = s^2\left(2c^2 \cos \Delta t_d \cos \omega t_d - s^2 \cos^2 \Delta t_d - s^2 + 2\right) \quad (3)$$

As an aside, we note that the usual transverse luminescence observation of quantum beats has the same form as the second term. It allows both extraction of $\Delta$, and may also be used for discrimination between quantum beats and polarisation interference[24] between two independent oscillators in situations, where the system is not well understood (not the case for the donors here). Quantum beating where the difference frequency is of the order 1 THz has been observed in semiconductor wells and dots using interband (near-IR) pumps[13,25–27]. Terahertz pulses can also induce other quantum coherent effects such as the Autler–Townes splitting[28], and ac Stark effect[29]. Here, we investigate a GHz beat induced by THz pump pulses, in this case using neutral donor orbital Rydberg states in silicon, thus extending the analogy with quantum beat experiments carried out for free atomic vapours and atoms in vacuum traps.

Our aim is to demonstrate the flexibility in state preparation provided a pair of pulses. The state (equation (3)) is described by the phases $\Omega t_p/\sqrt{2}$, $\Delta t_d$, $\omega t_d$, $\Delta t$, $\omega t$. Even if $\Delta$ and $\omega$ are fixed, the fact that $\Delta \ll \omega$ means we still have near complete freedom to choose $\Delta t_d$ independent of $\omega t_d$ and $\Delta t$ independent of $\omega t$. This is sufficient control to take an atom from the $1s$ ground state to most of the Hilbert space spanned by $1s$ and $2p_\pm$ at well-determined times (the value of $|\psi_0|^2$ can be varied from 0 to 1, but the maximum $|\psi_\pm|^2$ is 27/32). Complete control may be gained by allowing variable polarisation and amplitude of each pulse in the pair (see Methods).

**Theory of donor wavefunctions in silicon.** Silicon is an indirect gap semiconductor with 6 anisotropic conduction band minima lying near the X-points at $[\pm kx, 0, 0]$, $[0, \pm ky, 0]$, $[0, 0, \pm kz]$, where $k \approx 0.85 \ 2\pi/a$ and $a$ is the lattice constant. The KL

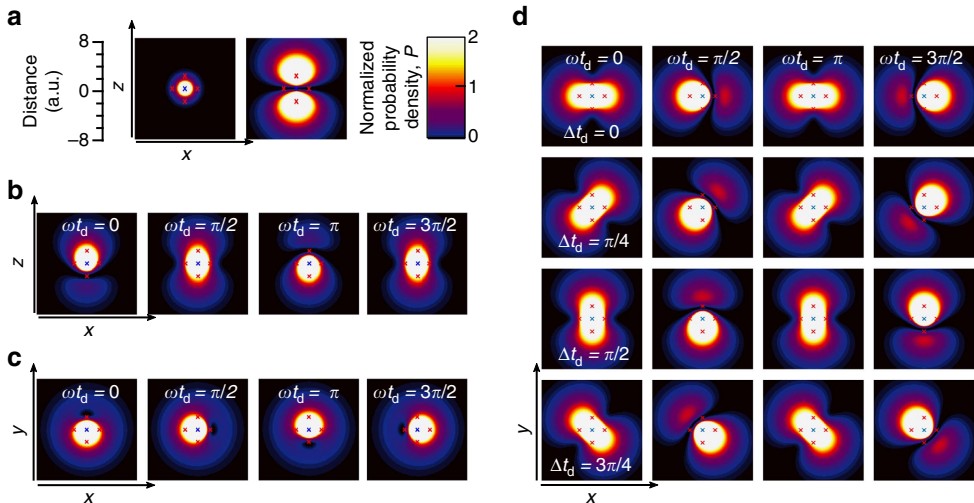

**Figure 1 | Control of hydrogen orbital wavefunctions using superposition states.** Heat maps of the probability density function are shown (**a,b** are cross-sections in the $x$–$z$ plane and **c,d** are in the $x$–$y$ plane). The probability density was normalized in each frame so that the probability of finding the electron in a region with $P \geq 1$ is 50%. The red contour thus encloses 50% of the integrated probability density. Blue crosses show the origin and red crosses are located at $(0, \pm 2a_0^*)$ and $(\pm 2a_0^*, 0)$ in-plane for reference where $a_0^*$ is the Bohr radius. (**a**) The ground state 1s and $2p_0$ excited state are examples of states which can be superposed using an intense light pulse. (**b**) An example of the 1s and $2p_0$ in an equal admixture, generated after applying a $z$-polarized $\pi/2$-pulse shown as a function of time. The superposition phase, $\varphi = \omega t$, where $\hbar\omega$ is the energy separation between the two basis states. (**c**) Time evolution of an equal admixture between the 1s and $2p_-$ states, which could be produced with a $\sigma_-$ circularly polarized light pulse shown as a function of time. (**d**) Admixture between the 1s, the $2p_-$ and the $2p_+$ following a short pulse with area $\Omega t_p = \sqrt{2\pi}/4$ (starting from 1s) shown as a function of time. The fast oscillation along the rows evolves with phase $\omega t$ and the slow rotation from row to row evolves with phase $\Delta t$, where $\hbar\omega$ is the average transition energy (9.47 THz) and $2\hbar\Delta$ is the difference between the excited states.

approach to the silicon donor[1,30] is to begin with the solutions of the Schrödinger equation for a single minimum at $k = 0$. The single valley 1s, $2p_0$ and $2p_\pm$ state wavefunctions are, to a good approximation, hydrogen-like with extent approximately equal to the effective Bohr radius $a_0^*$, but display a contraction along the valley axis due to the anisotropy in the effective mass (see Methods for more details). This anisotropy lifts the degeneracy between the $2p_0$ and $2p_\pm$.

The quantum defect of the donor atom and local strains produce a central cell correction (CCC) potential. The CCC makes the 1s state more tightly bound, that is, shrinks its extent. Although its precise form is unknown, the CCC has tetrahedral symmetry, which produces inter-valley mixing of the $1s^{x,y,z}$, and in Si:P (and most other shallow donors in silicon) one resulting multi-valley linear combination (called $1s(A_1)$ by KL and referred to here as $|0\rangle$ for brevity) has significantly lower energy than the other five:

$$|0\rangle \equiv |1s(A_1)\rangle = \frac{1}{\sqrt{3}}\left[|0^x\rangle + |0^y\rangle + |0^z\rangle\right] \quad (4)$$

where

$$|0^j\rangle \equiv \sqrt{2}\cos(kj)f_{1s}^j(\mathbf{r}) \quad (5)$$

is the ground state function translated from $k = 0$ to the $X$-points along the $j$ axis, where $j$ runs over $x$, $y$, $z$. There is a similar set of functions where the crystal part of the envelope is $\sin(kj)$, but as they only appear in the higher energy multi-valley components of 1s (called $1s(T_2)$ by KL), we ignore them here.

The excited states that have non-zero dipole moment overlap with $|0\rangle$, that is, those with odd parity, are almost unaffected by the CCC due to its short range. We let

$$|i^j\rangle \equiv \sqrt{2}\cos(kj)f_{2p_i}^j(\mathbf{r}) \quad (6)$$

for the excited state where $i$ and $j$ both run over $x$, $y$, $z$, so that for

example, $|x^z\rangle$ is the $2p_x$ state in the $z$-valley and so on. Again, there is a similar set of states with crystal part $\sin(kj)$ that are degenerate with our set $|i^j\rangle$, but they are dipole forbidden for transitions from $|0\rangle$ and again we ignore them here.

Since $k$ is large, the crystal part of the wavefunction oscillates rapidly compared with the slowly varying part of the envelope, $f^j(\mathbf{r})$, even in the case of the 1s state. This simplifies calculation of matrix elements of slowly varying operators; inter-valley matrix elements are negligible, and intra-valley matrix elements such as the dipole moment matrix elements may be approximated thus:

$$d_\pm = \langle 0^z|x|x^z\rangle = \iiint \mathrm{d}x\mathrm{d}y\mathrm{d}z\, 2\cos^2(kz)f_{1s}^z(\mathbf{r})xf_{2p_x}^z(\mathbf{r})$$

$$\approx \iiint \mathrm{d}x\mathrm{d}y\mathrm{d}z f_{1s}^z(\mathbf{r})xf_{2p_x}^z(\mathbf{r}) \quad (7)$$

in the example of a transition between 1s and the $2p_x$ state in the $z$-valley and so on. We find $d_\pm = 0.65a_0^*$. For light polarised with electric field along $\mathbf{e} = [e_x, e_y, e_z]$, the multi-valley $2p_\pm$ state that couples to $1s(A_1)$ is

$$|e\rangle = e_x\frac{|x^y\rangle + |x^z\rangle}{\sqrt{2}} + e_y\frac{|y^z\rangle + |y^x\rangle}{\sqrt{2}} + e_z\frac{|z^x\rangle + |z^y\rangle}{\sqrt{2}} \quad (8)$$

as shown in Fig. 2 for $\mathbf{e} = [001]$. The appropriate Rabi frequency for any $\mathbf{e}$ is $\hbar\Omega = \langle 0|\mathbf{r}|e\rangle \cdot \mathbf{F} = \sqrt{2/3}d_\pm F$, where $F$ is the electric field amplitude of the light wave.

In the presence of a magnetic field the static Hamiltonian is

$$H = H_0 + \mu_B^*\mathbf{B} \cdot \mathbf{L} + O\left(\mu_B^{*2}B^2/E_R^*\right) \quad (9)$$

where, $H_0$ is the zero field Hamiltonian that includes the anisotropic mass effects. $\mu_B^* = e\hbar/2m^* = 0.30$ meV T$^{-1}$ is the effective Bohr magneton and $E_R^* = 19.9$ meV is the effective Rydberg energy. In our experiment, $B \ll E_R^*/\mu_B^*$ ($\sim 66$ T) so the quadratic terms are negligible. The $2p_\pm$ excited states have their angular momentum, $\mathbf{L}$, quantized along the valley axis

(along ⟨001⟩ directions) and so for **B** parallel to [111] the linear $B$ term is the same for all valleys: the $2p_+$ states in all valleys obtain the same Zeeman energy $\hbar\Delta$, while the $2p_-$ states all obtain energy $-\hbar\Delta$. Figure 3a shows the calculated level spectrum in the low field limit with **B** along [111] using the Lanczos method described elsewhere[4]. These calculations

produce $\Delta/2\pi B = 42\,\text{GHz}\,\text{T}^{-1}$. Other field directions give rise to a more complex spectrum (see for example, ref. 4).

For the three levels produced by **B** along [111], we find (see Methods) that the time dependence of the amplitudes of the states may be taken directly from the preceding hydrogen theory equations (1–3), and Fig. 3b, which shows the time evolution according the equation (3), is appropriate for either situation. The 1s ground state of hydrogen is replaced by the silicon $1s(A_1)$ state, and the excited states by the silicon equivalents of the $2p_+$ and $2p_-$. The wavepacket components are

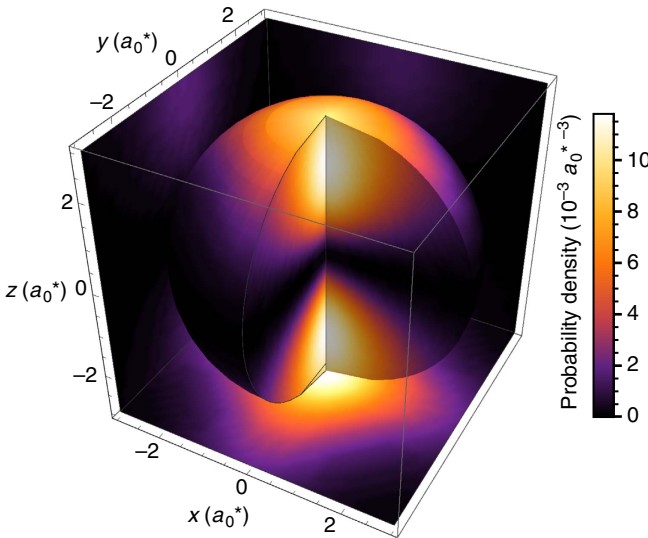

**Figure 2 | The silicon multi-valley $2p_\pm$ state.** A calculation of the silicon multi-valley $2p_\pm$ state appropriate for linearly polarized light in the $z$-direction, $\langle|\psi|^2\rangle_\text{period}(r) = \frac{1}{2}|f^x_{2p_z}(r)|^2 + \frac{1}{2}|f^y_{2p_z}(r)|^2$. The density is averaged over the quickly varying part of the envelope is shown on some surfaces near the donor. The length scale is in units of $a_0^* = 3\,\text{nm}$, the density scale is in units of $a_0^{*-3}$, and the edges of the cube are ⟨001⟩ directions. Note the cubic distortions of the wavefunction evident in the planar cut at the bottom.

$$\begin{bmatrix}\langle 0^x|\psi\rangle\\\langle y^x|\psi\rangle\\\langle z^x|\psi\rangle\\\langle 0^y|\psi\rangle\\\langle z^y|\psi\rangle\\\langle x^y|\psi\rangle\\\langle 0^z|\psi\rangle\\\langle x^z|\psi\rangle\\\langle y^z|\psi\rangle\end{bmatrix} = \frac{i}{\sqrt2}\begin{bmatrix}e_0 & 0 & 0\\0 & e_z & e_y\\0 & -e_y & e_z\\e_0 & 0 & 0\\0 & e_x & e_z\\0 & -e_z & e_x\\e_0 & 0 & 0\\0 & e_y & e_x\\0 & -e_x & e_y\end{bmatrix}\begin{bmatrix}1 & 0 & 0\\0 & \frac{-1}{\sqrt2} & \frac{1}{\sqrt2}\\0 & \frac{-i}{\sqrt2} & \frac{i}{\sqrt2}\end{bmatrix}\begin{bmatrix}\psi_0\\\psi_+\\\psi_-\end{bmatrix}$$

(10)

where, $e_0 = -i\sqrt{2/3}$. Figure 3c shows the evolution of the density after a pulse polarised along [11–2] appropriate for our experiment. The density has been averaged over a volume $(2\pi/k)^3$ corresponding to one oscillation period of the crystal part of the wavefunction,

$$\langle|\psi|^2\rangle_\text{period}(\mathbf{r}) = \sum_j\left|\sum_\alpha\langle\alpha^j|\psi\rangle f^j_\alpha(\mathbf{r})\right|^2$$

(11)

Figure 1 shows a hydrogen atom, and the Si:P donor Fig. 3c has a larger scale as already mentioned but is otherwise equivalent—differing essentially by some cubic distortions (the [111] plane shown has hexagonal symmetry) and a relatively more compact

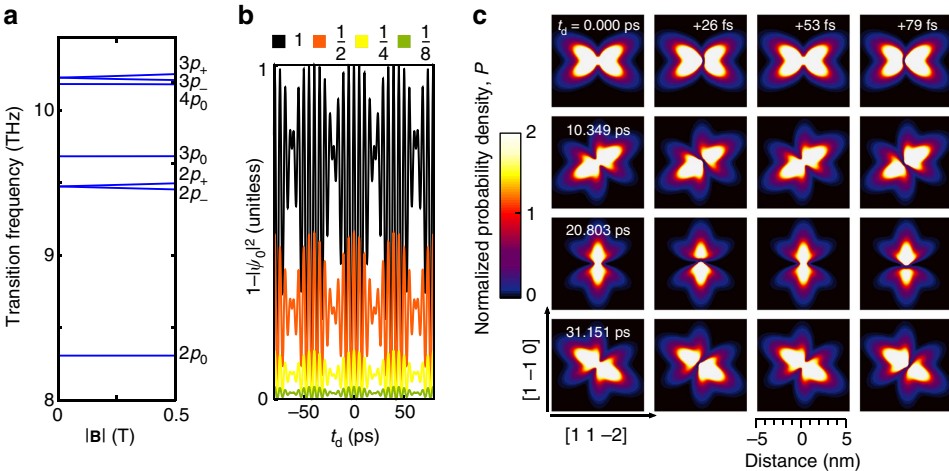

**Figure 3 | Time evolution of multi-valley donor states in silicon.** Evolution through time of a multi-valley superposition of the $1s(A_1)$ and $2p_\pm$ states in Si:P for a magnetic field direction of [111]. (**a**) The calculated level spectrum in the low field limit and **B** along [111]. (**b**) The absorption equation (3) (no dephasing) as a function of delay for a pair of linearly polarized pulses of equal area $\Omega t_p = \alpha\theta_0$ each, where $\theta_0 = \sqrt2\pi/4$ for different values of $\alpha$ according to the legend ($\alpha = 1$ corresponds to both pulses having area equal to the illustration in Figs 1d and 3c, and $\alpha = 1/8$ corresponds to the experimental condition in Figs 4,6 and 7 below). (For clarity, the fast oscillation at 9.47 THz has been aliased down by a factor 50 by under sampling). (**c**) The probability density shown in the plane perpendicular to the field. The superposition is initiated at $t = 0$ by a delta-function pulse polarized along [11–2], reducing the probability of occupation of the ground state to 50% which may be produced by a uniform illumination of pulse of energy of 10 μJ duration 10 ps, and spot size of 1 cm (equivalent to the unfocussed laser beam). The density has been spatially averaged over the quickly varying oscillations due to valley interference. The probability density was normalized in each frame so that the probability of finding the electron in a region with $P \geq 1$ is 50%. Thus the red contour encloses 50% of the integrated probability density. The labelled times $t_d$ are shown in ps, equivalent to the phases shown in Fig. 1, that is, $\Delta t_d = n\pi/2$ and $\omega t_d = m\pi/2$, where $n$ and $m$ index rows and columns, respectively, and range from 0 to 3. In this example, the Zeeman splitting is $2\Delta = 24.2\,\text{GHz}$, which occurs for $|\mathbf{B}| = 0.285\,\text{T}$.

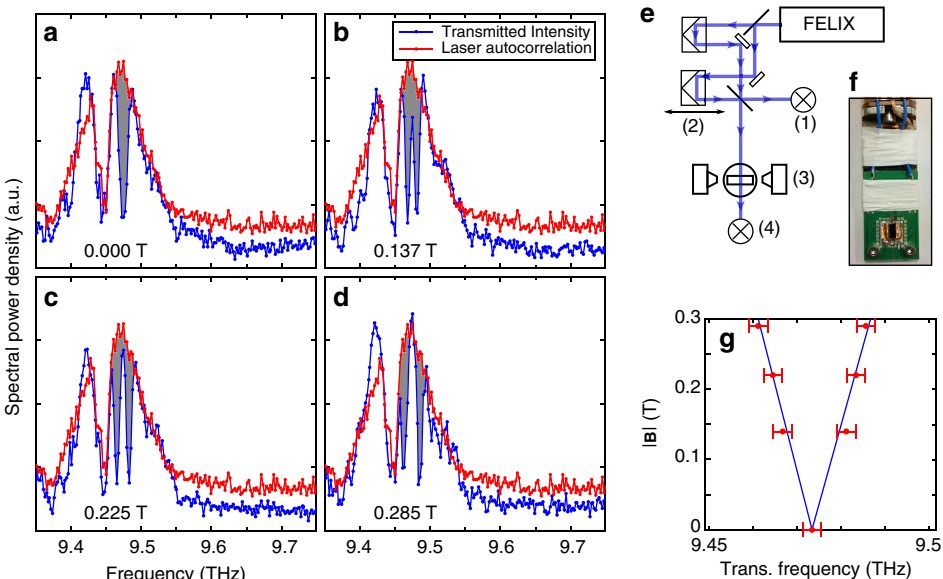

**Figure 4 | Frequency domain quantum beats in transmission.** (**a**–**d**) Transmission spectra of the sample under varying magnetic field amplitudes, noted in the figure. Each panel shows the FT of the time-domain Ramsey fringes in the transmitted light (single channel, that is, unratioed with any reference) for different applied magnetic fields (blue points). The linear autocorrelation of the laser, that is, the FT of the fringes taken simultaneously with a reference detector at the other exit port of the interferometer (red points) is clearly about 0.1 THz wide, and there is evidence of absorption lines from residual water vapour in the beam path (for example, at 9.45 THz). The absorption peaks due to the sample are highlighted with a grey fill. (**e**) Optical layout schematic of the experiment: (1) autocorrelation reference detector, (2) computer controlled delay stage, (3) sample cryostat between electromagnet poles, (4) primary transmission measurement. (**f**) Photograph of the sample mounted upon the cold-finger, supported strain-free by copper foil capacitor plates. There is an aperture behind the sample to allow transmission. (**g**) Measured sample transmission line-centres (red points) taken from the shaded features in the main panels (error bars indicate the datapoint spacing in the spectrum). The blue lines show the theory for the 1s(A₁) to 2p± transitions from Fig. 3b.

ground state wavefunction due to the CCC. The change in ground state size due to the CCC is behind the relatively low contrast in changes along rows in Fig. 3c compared to Fig. 1d.

**Evolution of superposition wavefunctions in silicon**. The first term in equation (3) is normally very difficult to access, but for Si:P $\omega$ is in the THz region of the spectrum, and thus we are easily able to produce control over $\omega t_d$ with high precision (0.05 radians was our instrument limit) using ordinary stepper motors. We are therefore not restricted to investigation solely of dynamics on the scale of $\Delta t_d$, and so are able to probe the beat directly via the first term in equation (3).

The parameter-space is large so we restrict our investigation to the $t_d$ dependence, which exhibits variation in two of the five phases in equation (2). We chose to use weak pulses ($\Omega t_p \sim \pi/20$) to avoid multi-photon effects, since we are anyway interested in the $s^2$ term in equation (3) rather than the $s^4$ terms. The latter are more susceptible to intensity jitter and were unobserved. Reducing $s$ clearly changes only the amplitudes of the $t_d$-dependent terms, equation (3) (Fig. 3b).

**Experiment**. The transmitted light through the sample was measured as a function of the pulse delay for different magnetic field strengths. The FT of equation (3) (with respect to $t_d$) contains a doublet at $\omega \pm \Delta$ (as well as a zero frequency term and a term at $2\Delta$) and Fig. 4 shows the FTs of the experimental interferograms in which the doublet is clearly observed, with $\Delta/2\pi B = 42$ GHz T$^{-1}$ as expected from the level splitting calculated in Fig. 3b.

These experiments are similar to a variety of pulsed-laser spectroscopies where a short optical pulse is used to excite an oscillation, and correlations between copies of these oscillations

are measured by taking advantage of interference effects. Our technique is distinct primarily in that the free-electron laser (FEL) source used allows THz pulses with a selective, narrower band corresponding to a few hundred coherent field oscillations. As for other experiments of this type, and in contrast to incoherent frequency domain spectroscopies such as FTIR, the interference is most intuitively understood in terms of the time-domain behaviour: the second pulse interferes with the long-lived excitation produced by the first pulse. The time-domain interference pattern is the autocorrelation of the sample impulse response, so its Fourier transform is its frequency spectrum.

The characteristic resolving power of our time-domain interferometer was tested by measuring the transmission fringes induced by an optically polished slab of high purity float-zone silicon at room temperature, shown in Fig. 5. The results were compared with the transmission measured by high-resolution FTIR with resolution 0.01 cm$^{-1}$ (0.0003 THz), and excellent agreement was produced.

The features in the transmission are superimposed on a background due to the system response (red data on Fig. 4), which includes the bandwidth-limited laser spectrum[31] and an absorption line from some residual water vapour in the beam path, containing a few cm of air between the evacuated optical setup and the cryostat. This means that the raw time-domain data include oscillations with several periodicities and are therefore not interpretable by direct inspection. It is preferable to expose the quantum level splittings by examination of the proportion of excited atoms within the sample rather than the transmitted field pattern at a point far from the sample. This is for the simple reason that the transmitted light spectrum is the product of the transmission spectrum with the system response spectrum (as in Fig. 4), which contains a complicated background, while the product of the absorption spectrum and the system response

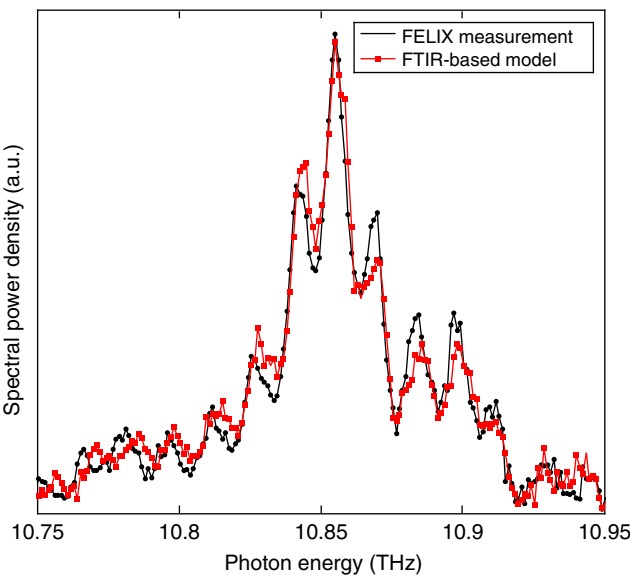

**Figure 5 | Calibration of the experimental setup using a Si optical flat.**
Red: A reference transmission measurement of the optical flat was obtained using a high-resolution FTIR spectrometer, and multiplied by the system response (red data on Fig. 4) to give a model of the expected transmission profile of the FELIX measurement. Black: FT of the FELIX transmission interferogram for the optical flat. Good agreement is seen within the noise, which is limited by the beam-time available.

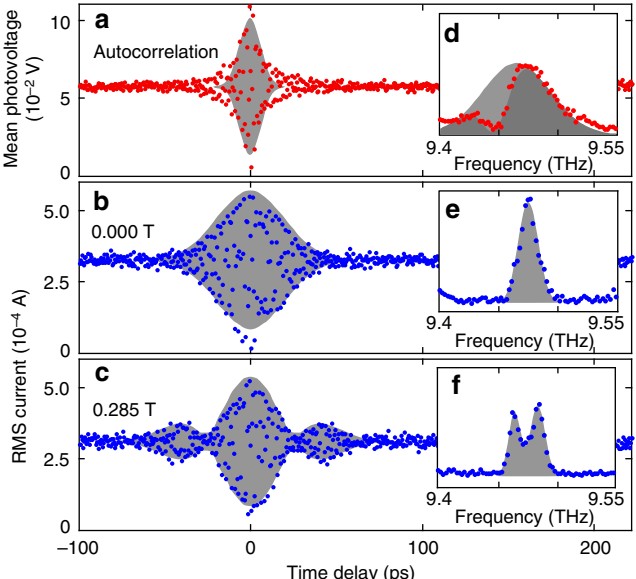

**Figure 6 | Quantum beats in absorption.** (**a**) Laser pulse autocorrelation interferogram from detector at (1) in Fig. 4e, showing the overlap of the laser pulses. (**b,c**) Absorption (PTIS) interferogram for different applied magnetic fields (noted in the figure). (**d–f**) FTs of the interferograms shown in (**a–c**). The grey envelopes in the frequency domain are fits with the FT of equation (3) convolved with a Gaussian lineshape (in the case of the laser autocorrelation we included a residual water vapour line at 9.45 THz). The inverse FTs of these fits are shown in grey on the time-domain **a–c**. In the case of (**d**) (the autocorrelation), the laser lineshape is shown shaded light grey; a darker shade is used to show the effect of subtracting the fitted water line.

spectrum is background-free, that is, it contains just the atomic transition features. One method, contactless photothermal ionization spectroscopy (PTIS), relies on a small degree of phonon-induced ionisation from the excited state, which generates a small number of free charge carriers. These conduction electrons may be detected via the AC dielectric properties of the sample[32,33]. The number of donors ionized is proportional to the average population in the excited states[21].

We show in Fig. 6 a set of contactless PTIS interferograms which display a beat in the excited state probability. Again, we kept the intensity low ($\Omega t_p \sim \pi/20$) to avoid multi-photon effects. When there is no applied field and the excited states are degenerate ($\Delta = 0$), a single lobe is observed in the interferogram at zero time delay, which decays on a time-scale determined by the inverse of the inhomogeneous line width. The single lobe becomes a characteristic beat pattern upon the application of a magnetic field. Figure 6d–f also shows the FTs of these interferograms, the peak positions of which correlate directly to the absorptions observed in Fig. 4 and predicted by equation (3). We performed fits in the frequency domain using the FT of equation (3) convolved with a gaussian broadening, and the results are shown in grey. The inverse FTs of the fits are also shown in the time domain. The beat period is clearly controllable with the magnetic field, Fig. 7, hence the beating behaviour is induced by the energy level splitting in the atoms according to equations (2) and (3).

Figure 6a shows the laser autocorrelation interferogram aquired simultaneously with the zero field PTIS. It dies off much more rapidly than the PTIS (in spite of a slight lengthening in the time-domain due to the sharp water vapour line); in other words we have shown that the coherent excitation in the sample is significantly longer lived than the light pulse duration. Our experiment of Fig. 6c demonstrates control over the relative phase and amplitude of the $2p_+$ and $2p_-$ states using a pair of identical linearly $x$-polarized pulses, which we illustrate by comparison to Fig. 3b. The antinodes of the beat pattern in

Fig. 6c are points at which the second pulse arrives when both excited states are in phase or 180° out of phase with each other (first and third rows of Figs 1d and 2a); they are equally doubled (maxima in the rapid oscillations at the average frequency) or destroyed (minima). The nodes in Fig. 6c, on the other hand, are points where the second pulse arrives when both excited states are 90° out of phase (second and fourth rows of Figs 1d and 3a); the second pulse transfers amplitude from one component to the other with no change in the total excited state population.

## Discussion
We have used a coherent time-domain analogue of FTIR to examine the magnetic field split $1s$–$2p_\pm$ transition in Si:P. We have made a calculation for donor wavefunctions in Si in small **B** along the [111] crystal axis; simultaneous pumping of two Zeeman split transitions induces beating, thus demonstrating control over a three-state admixture. This control provides an additional tuning parameter, that is, the relative admixture of the three states, for production of Si:donor-based quantum logic gates. Utilization of the orbital motion in the $x$–$y$ plane could have great advantages for surface code networks[14–17]. Suppose there are physical (spin) qubits positioned at the red cross positions in Fig. 1, and that we position a fiducial donor qubit of a different species (P in our case) at the centre; the radius of the wavefunction for the spin qubits is much less than the separation between them so as to avoid direct overlap in the ground state. The central donor can be excited to the $2p_+$ state by a THz pulse, while the surrounding qubits remain unexcited because the associated Rydberg levels are at different energies. The excited donor electron now spends time in the vicinity of the logical

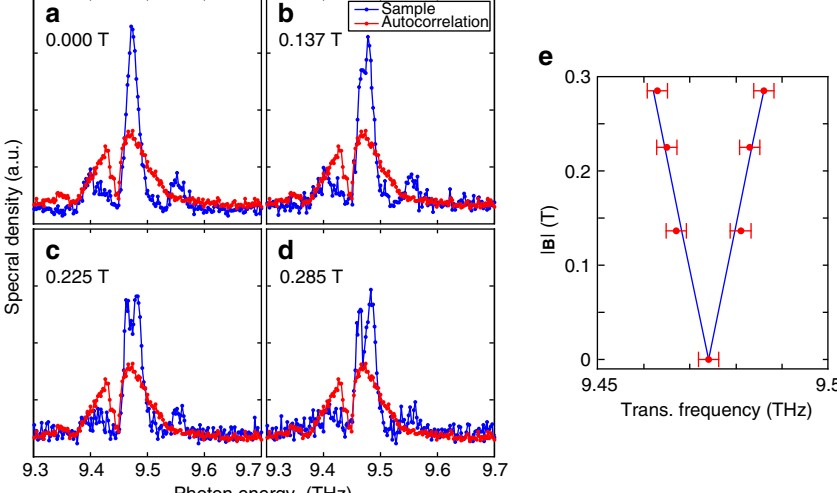

**Figure 7 | Splitting of the 1s to 2p$_\pm$ transition measured in absorption.** (**a–d**) Spectra of the 1s(A$_1$) to 2p$_\pm$ transition, measured using absorption (PTIS) for a variety of magnetic fields noted in the figure. The data have lower frequency resolution and lower statistical power from averaging than those of Fig. 6 to provide a better coverage of the magnetic field degree of freedom. (**e**) Extracted peak locations from (**a–d**) (red points) with error bars showing $\pm$ one datapoint, compared to the expected variation (blue trace), at the four different magnetic field amplitudes shown in **a–d**.

qubits during which it experiences exchange interaction, and acquires a phase depending on their state. Projecting the fiducial qubit back onto $z$ at an appropriate time and then observing it is equivalent to evaluating a weighted sum of surrounding qubits, which is the required operation for error detection. The strength of the exchange interaction available in the excited state has been considered by ref. 34. Controlling the orbital trajectory by controlling superpositions of $2p_+$ and $2p_-$ gives additional control, which could allow a bigger variety of fiducial measurement strategies. Finally, we note that our THz quantum beat spectroscopy technique opens up the possibility of high-resolution spectroscopies for GHz splittings of excited states.

## Methods

**Experimental methods.** The magnetic field was provided by a Newport Type C water cooled electromagnet, which was calibrated using a Hall probe. The maximum field measured was 0.285 T at an applied current of 2 A. Our Si:P sample was ordinary commercial float-zone grown silicon (natural isotope composition), with growth direction <110>. The dopant density was $2 \times 10^4$ cm$^{-3}$. It was mounted in a an Oxford Instruments Microstat, in vacuum, cooled by liquid helium to ∼10 K. The cryostat containing the sample was placed between the poles of the magnet, such that the sample surface normal was approximately perpendicular to the magnetic field, which was parallel to a <111> direction.

The sample was suspended strain-free, between a pair of capacitor plates formed using pockets of copper foil which did not wrap tightly around the sample, and a PCB. No DC electrical conduction was measurable through the system either at room temperature or under cryogenic conditions. The electronic measurement was obtained by supplying a 1 V peak–peak sine wave (15 MHz) generated using a HAMES 50 MHz HMF2550 arbitrary waveform generator, and measuring the supplied current to the circuit. The current was amplified using a FEMTO DHCPA-100 transimpedance amplifier with $10^2$ V A$^{-1}$ gain and 200 MHz bandwidth, and measured through a National Instruments PXIe-5162 oscilloscope. Figure 4 shows the r.m.s. current for $|\mathbf{B}| = 0$ and 0.285 T, a fuller field dependence (at taken with lower resolution due to laser beam-time restraints) is shown in Fig. 7.

Laser pulses from the Dutch terahertz Free-Electron Laser FELIX were passed through a variable delay Mach–Zender interferometer arrangement. Figure 4e shows an optical layout diagram of the experiment, with the location of each detector and the sample. A pellicle beam-splitter was used to split the FELIX beam into a pair of equal Ramsey pulses, as in ref. 21, and recombined using a pellicle beam-splitter in a Mach–Zender arrangement. The laser beams were linearly polarized perpendicular to the plane of the bench, and normally incident on the sample with electric field perpendicular to the static magnetic field (Voigt geometry). To measure an interferogram, the optical path difference between the pulses was varied using a computer controlled delay stage. At each value of path difference, the electronic waveform from each detector was recorded over one FELIX macropulse (∼110 micropulses). Datapoints were obtained from

the mean or RMS of the waveform over the stable part of the macropulse, to minimize noise.

The laser was tuned to resonance with the 1s(A$_1$)–2p$_\pm$ orbital transition, that is, optical frequency 9.48 THz. The FELIX pulses had a spectral linewidth FWHM 0.47% and are bandwidth limited, corresponding to a temporal power FWHM of 9.8 ps. The measured micropulse energy at the sample per beam was 30–50 μJ before the cryostat window. The light intensity transmitted through the sample was measured using a liquid helium cooled Ga:Ge detector. The laser autocorrelation spectrum was taken simultaneously using wasted light at the other output of the downstream beam-splitter (Fig. 4e) using a pyroelectric detector. The beam-splitters are not 50:50 and the fringe contrast of the autocorrelation signal is poorer than for the output directed towards the sample, but it has the same temporal profile.

Optical signals were averaged over the FELIX macropulse, and the r.m.s. was measured for the electrically detected signal. The interferograms of Fig. 6 are unprocessed. Further data processing was applied to the interferograms before their Fourier transform was taken to reduce machine precision errors, including: running median background subtraction, normalization and correction for systematic errors along the time axis (on the order of 0.02%). To minimize data collection time, under sampling was employed such that the Nyquist frequency was 1 THz, and high frequency information was extracted by unfolding the spectrum in a standard manner.

The characteristic resolving power of our interferometer was tested by measuring the Fabry–Perot transmission fringes induced by an optically polished slab of high purity float-zone silicon at room temperature. The slab was of thickness 3.05 mm and a calibration standard was set by transmission measurement in a high-resolution FTIR IFS125HR with resolution 0.01 cm$^{-1}$ (0.0003 THz). We compare the FT of the transmission interferogram of the FELIX pulse through the slab, with the product of the FELIX autocorrelation spectrum and the reference spectrum in Fig. 5. The results compare favourably, so we set an upper bound for the resolution at 0.0146 THz, which is given by the peak spacing of the fringes. The lower bound is 0.0003 THz, limited by the length of the delay line[34].

The fidelity of the fringes in Fig. 6 was calculated as (median − minimum)/ median in the centre 10% of the interferogram data set. It was 95% for 0.000 T and 82% for 0.285 T. The readout fidelity in these measurements is limited most strongly by the under sampling of the interferogram, and a more densely sampled measurement would produce a stronger fidelity.

**Theory of three-level quantum optics.** The state vector **ψ** with components $|2p_-\rangle$, $|1s\rangle$, $|2p_+\rangle$, and with amplitudes $\psi_-$, $\psi_0$, $\psi_+$,

$$|\psi\rangle = \psi_-|2p_-\rangle + \psi_0|1s\rangle + \psi_+|2p_+\rangle, \qquad (12)$$

evolves according to the time-dependent Schrödinger equation $H\psi = i\hbar\,\partial\psi/\partial t$. The off-diagonal matrix elements of the perturbation for a wave of frequency $\omega$ and phase $\theta$, polarized along $x$ is $V = \hbar\Omega\cos(\omega t - \theta)$, where the Rabi frequency is $\Omega = \mu_x F_x/\hbar$, the electric field amplitude is $F_x$, and $\mu_x$ is the dipole moment, and the Hamiltonian for a basis order of $2p_-$, $1s$, $2p_+$ is

$$H = \begin{bmatrix} E_- & V & 0 \\ V & E_0 & V \\ 0 & V & E_+ \end{bmatrix} \qquad (13)$$

**Table 1 | KL wavefunctions and parameter.**

| State, $\alpha$ | Single valley components $f^j_\alpha(\mathbf{r})$ for $j = z$ | $L/a^*_0$ | $\Gamma$ |
|---|---|---|---|
| 1s | $f^z_{1s}(\mathbf{r}) = (\pi L^3)^{-1/2}\Gamma^{-1/4}\exp(-L^{-1}\sqrt{x^2+y^2+z^2/\Gamma})$ | 0.78 | 0.33 |
| $2p_0$ | $f^z_{2p_z}(\mathbf{r}) = (32\pi L^5)^{-1/2}\Gamma^{-3/4}z\exp(-\frac{1}{2}L^{-1}\sqrt{x^2+y^2+z^2/\Gamma})$ | 0.59 | 0.37 |
| $2p_\pm$ | $f^z_{2p_x}(\mathbf{r}) = (32\pi L^5)^{-1/2}\Gamma^{-1/4}x\exp(-\frac{1}{2}L^{-1}\sqrt{x^2+y^2+z^2/\Gamma})$ | 0.88 | 0.38 |

Single valley KL variational wavefunctions and parameters[30], in which $m_t = 0.1905m_0$, $m_l = 0.9163m_0$, $\varepsilon_r = 11.4$, and consequently the renormalized Bohr radius $a^*_0 = 3.17$ nm and $\gamma = m_t/m_l = 0.208$. In $f^j_\alpha(\mathbf{r})$ the superscript indicates the valley axis, and $2p_0 = 2p_z$ and $2p_\pm = -(2p_x \pm i2p_y)/\sqrt{2}$. $f^z_{2p_x}(x,y,z) = f^z_{2p_x}(y,x,z)$.

We make a transformation $\boldsymbol{\psi} = U\boldsymbol{\chi}$ to a rotating frame in which

$$|\chi\rangle = \chi_-|2p_-\rangle + \chi_0|1s\rangle + \chi_+|2p_+\rangle \quad (14)$$

and

$$U = \exp\left(-\frac{iE_0 t}{\hbar}\right)\begin{bmatrix} \exp(-i\omega t) & 0 & 0 \\ 0 & 1 & 0 \\ 0 & 0 & \exp(-i\omega t) \end{bmatrix} \quad (15)$$

The time-dependent Schrödinger equation in the rotating frame is $\partial\boldsymbol{\chi}/\partial t = -iW\boldsymbol{\chi}$, where

$$W = \frac{1}{2}\begin{bmatrix} -2\Delta & \Omega e^{i\theta} & 0 \\ \Omega e^{-i\theta} & 0 & \Omega e^{-i\theta} \\ 0 & \Omega e^{i\theta} & 2\Delta \end{bmatrix} \quad (16)$$

after use of the rotating wave approximation. Here, $2\hbar\Delta = E_+ - E_-$ is the Zeeman splitting of the excited state. The drive frequency is assumed to be half way between the 1s to $2p_\pm$ transitions, but for short pulses this restriction is unimportant. The result of integrating $\partial\boldsymbol{\chi}/\partial t$ is $\boldsymbol{\chi}_{\text{final}} = T(t)\boldsymbol{\chi}_{\text{initial}}$ where

$$T(t) = \exp(-iWt) = \mathbf{I} - i\sin(\Omega't)W/\Omega' - [1-\cos(\Omega't)]W^2/\Omega'^2 \quad (17)$$

and $\Omega'^2 = \Omega^2/2 + \Delta^2$. The result for a delta-function pulse (pulse duration $t_p \ll \Delta^{-1}$) on an atom starting in the ground state is

$$\chi_0 = c, \chi_\pm = -is/\sqrt{2} \quad (18)$$

where, $c = \cos(\Omega t_p/\sqrt{2})$, $s = \sin(\Omega t_p/\sqrt{2})$. In the dark, a time $t_d$ after the pulse, the result is

$$\chi_0 = c, \chi_\pm = -ise^{\pm i\Delta t_d}/\sqrt{2} \quad (19)$$

and returning to the laboratory frame, $\boldsymbol{\psi} = U\boldsymbol{\chi}$ produces equation (1) (ignoring the global phase factor in front of $U$). The effect of a second, identical pulse (arriving at time $t_d$) and a further wait in the dark of time $t$ is equation (2). The state is described by the phases $\Omega t_p/\sqrt{2}$, $\Delta t_d$, $\omega t_d$, $\Delta t$, $\omega t$. Even if $\Delta$ and $\omega$ are fixed, the fact that $\Delta \ll \omega$ means we still have near complete freedom to choose $\Delta t_d$ independent of $\omega t_d$ and $\Delta t$ independent of $\omega t$.

**Theory of donor wavefunctions in silicon.** The KL[1] multi-valley wavefunction (excluding the atomic part of the Bloch function) is

$$\psi(\mathbf{r}) = \sum_{v=1}^{6} \beta_v \exp(i\mathbf{k}_v \cdot \mathbf{r})f^v(\mathbf{r}) \quad (20)$$

where for valley $v$, $\mathbf{k}_v$ is the wavevector at the valley minimum, thus $\mathbf{k}_v.\mathbf{r}$ runs over $\pm kx$, $\pm ky$, $\pm kz$, where $k = 0.85\ 2\pi/a$ and $a$ is the lattice constant. The single-valley envelope function $f^v(\mathbf{r})$ is the same for valleys in opposite directions.

As mentioned in the main text, the single valley odd parity excited state wavefunctions are, to a very good approximation, hydrogen-like with a length scale, $L$, that is approximately equal to the effective Bohr radius $a^*_0$, and with a contraction along the valley axis by a factor $\sqrt{\Gamma}$, where $\Gamma \approx \gamma = m_t/m_l$ the ratio of effective masses transverse and longitudinally along the valley axis (Table 1). This anisotropy lifts the degeneracy between the $2p_0$ and $2p_\pm$. The wavefunctions may be obtained from a Lanczos procedure[4] or variational calculations[1,30]. The $z$-valley variational envelope functions are given in Table 1.

For a superposition the density is

$$|\psi|^2(\mathbf{r}) = \left|\sum_j \sqrt{2}\cos(kj)\sum_\alpha \langle\alpha^j|\psi\rangle f^j_\alpha(\mathbf{r})\right|^2 \quad (21)$$

where $f^j_\alpha(\mathbf{r})$ is the spatial profile of state $|\alpha^j\rangle$. Neglecting the inter-valley terms

$$|\psi|^2(\mathbf{r}) \approx \sum_j 2\cos^2(kj)\left|\sum_\alpha \langle\alpha^j|\psi\rangle f^j_\alpha(\mathbf{r})\right|^2 \quad (22)$$

Averaging over a volume $(2\pi/k)^3$ corresponding to one oscillation period of the crystal part of the wavefunction:

$$\langle|\psi|^2\rangle_{\text{period}}(\mathbf{r}) = \sum_j \left|\sum_\alpha \langle\alpha^j|\psi\rangle f^j_\alpha(\mathbf{r})\right|^2 \quad (23)$$

The $|0\rangle$ ground state contains contributions from all valleys, and in general has non-zero dipole moment matrix element with the excited states in more than one valley. The dipole moment matrix elements are of the form $\langle 0^i|\beta|\alpha^j\rangle$ with $i, j, \alpha, \beta$ running over $x, y, z$. From symmetry arguments

if $\alpha = j (2p_0$ states) then $\langle 0^i|\beta|\alpha^\alpha\rangle = d_0$ if $\alpha = \beta = i$ and zero otherwise (24)

if $a \neq j (2p_\pm$ states) then $\langle 0^i|\beta|\alpha^j\rangle = d_\pm$ if $\alpha = \beta$ and $i = j$ and zero otherwise (25)

For an electric field $\mathbf{F} = F[e_x, e_y, e_z]$, where $[e_x, e_y, e_z]$ is the normalized polarization direction and $[e_x, e_y, e_z].\mathbf{F} = |\mathbf{F}| = F$, we can construct a normalized excited state $|e\rangle$ whose dipole moment matrix element $\langle 0|\mathbf{r}|e\rangle$ is parallel to $\mathbf{F}$. In the case of the $2p_0$:

$$|e\rangle = e_x|x^x\rangle + e_y|y^y\rangle + e_z|z^z\rangle \quad (26)$$

$$\Rightarrow \langle 0|\mathbf{r}|e\rangle \cdot \mathbf{F} = \langle 0|[x,y,z]|e\rangle \cdot \mathbf{F} = \frac{1}{\sqrt{3}}[e_x d_0, e_y d_0, e_z d_0] \cdot \mathbf{F} = \frac{1}{\sqrt{3}}d_0 F \quad (27)$$

so the Rabi frequency is $\Omega = d_0 F/\sqrt{3}\hbar$, while for the $2p_\pm$ it is

$$|e\rangle = e_x\frac{|x^z\rangle+|x^y\rangle}{\sqrt{2}} + e_y\frac{|y^x\rangle+|y^z\rangle}{\sqrt{2}} + e_z\frac{|z^y\rangle+|z^x\rangle}{\sqrt{2}} \quad (28)$$

$$\Rightarrow \langle 0|\mathbf{r}|e\rangle \cdot \mathbf{F} = \langle 0|[x,y,z]|e\rangle \cdot \mathbf{F} = \frac{1}{\sqrt{6}}[2e_x d_\pm, 2e_y d_\pm, 2e_z d_\pm] \cdot \mathbf{F} = \sqrt{\frac{2}{3}}d_\pm F \quad (29)$$

so the Rabi frequency is $\Omega = \sqrt{2}d_\pm F/\sqrt{3}\hbar$. It is easy to show that $\langle 0|\mathbf{r}|e'\rangle \cdot \mathbf{F} = 0$ for all other excited states $|e'\rangle$ such that $\langle e|e'\rangle = 0$, that is, they are not involved in dipole excitation and can be ignored. Figure 2 shows the $2p_\pm$ excited state superposition appropriate for $z$-polarized light, that is,:

$$[e_x, e_y, e_z] = [001] \therefore \langle|\psi|^2\rangle_{\text{period}}(\mathbf{r}) = \frac{1}{2}\left|f^x_{2p_z}(\mathbf{r})\right|^2 + \frac{1}{2}\left|f^y_{2p_z}(\mathbf{r})\right|^2 \quad (30)$$

For $\mathbf{B}$ parallel to [111], all the $2p_+$ states in all the valleys obtain the same Zeeman energy $\hbar\Delta$, while the $2p_-$ states all obtain energy $-\hbar\Delta$, so to find the time evolution we resolve the superposition into its $2p_+$ and $2p_-$ components. We therefore change to a basis of $2p_+$ and $2p_-$ states in each valley:

$$|+^z\rangle = -\frac{|x^z\rangle+i|y^z\rangle}{\sqrt{2}}, |-^z\rangle = \frac{|x^z\rangle-i|y^z\rangle}{\sqrt{2}} \quad (31)$$

and cyclic permutations of $x, y, z$, and resolve the excited state into two components $|+\rangle$ and $|-\rangle$:

$$|\pm\rangle = \sum_j |\pm^j\rangle\langle\pm^j|e\rangle \quad (32)$$

At this point, all the three-level quantum optics theory from the preceding section now follows as before, so after a delta-function pulse and a wait in the dark, $|\psi\rangle = \psi_0|0\rangle + \psi_+|+\rangle + \psi_-|-\rangle$, where the amplitudes are given by $\psi_0 = c$, $\psi_\pm = -ise^{-i(\omega\mp\Delta)t_d}$. To find the density, we require the amplitudes $\langle j^k|\psi\rangle$. It is convenient to change basis again

$$|X\rangle = \frac{|-\rangle-|+\rangle}{\sqrt{2}}, |Y\rangle = i\frac{|-\rangle+|+\rangle}{\sqrt{2}} \quad (33)$$

so that $|\psi\rangle = \psi_0|0\rangle + \psi_X|X\rangle + \psi_Y|Y\rangle$,

$$\begin{bmatrix} \langle 0|\psi\rangle \\ \langle X|\psi\rangle \\ \langle Y|\psi\rangle \end{bmatrix} = \begin{bmatrix} \psi_0 \\ \psi_X \\ \psi_Y \end{bmatrix} = \begin{bmatrix} 1 & 0 & 0 \\ 0 & -\frac{1}{\sqrt{2}} & \frac{1}{\sqrt{2}} \\ 0 & \frac{-i}{\sqrt{2}} & \frac{i}{\sqrt{2}} \end{bmatrix}\begin{bmatrix} \psi_0 \\ \psi_+ \\ \psi_- \end{bmatrix} = \begin{bmatrix} c \\ -se^{-i\omega t_d}\sin(\Delta t_d) \\ -se^{-i\omega t_d}\cos(\Delta t_d) \end{bmatrix} \quad (34)$$

and now

$$
\begin{bmatrix}
\langle 0^x|\psi\rangle \\
\langle y^x|\psi\rangle \\
\langle z^x|\psi\rangle \\
\langle 0^y|\psi\rangle \\
\langle z^y|\psi\rangle \\
\langle x^y|\psi\rangle \\
\langle 0^z|\psi\rangle \\
\langle x^z|\psi\rangle \\
\langle y^z|\psi\rangle
\end{bmatrix}
=
\frac{i}{\sqrt{2}}
\begin{bmatrix}
e_0 & 0 & 0 \\
0 & e_z & e_y \\
0 & -e_y & e_z \\
e_0 & 0 & 0 \\
0 & e_x & e_z \\
0 & -e_z & e_x \\
e_0 & 0 & 0 \\
0 & e_y & e_x \\
0 & -e_x & e_y
\end{bmatrix}
\begin{bmatrix}
\psi_0 \\
\psi_X \\
\psi_Y
\end{bmatrix}
\tag{35}
$$

where, $e_0 = -i\sqrt{2/3}$. Figure 3a shows the evolution of $\langle|\psi|^2\rangle_{period}$ after a pulse polarized along [11–2] appropriate for our experiment.

**Data availability.** The data for Figs 3a,4a–d,5,6a–f and 7a–d are available online[35]. Data in more verbose formats are available upon request to the corresponding author.

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

## Acknowledgements

We acknowledge financial support from the UK Engineering and Physical Sciences Research Council [COMPASSS/ADDRFSS, Grant No. EP/M009564/1], and from the research programme of the Foundation for 'Fundamental Research on Matter' (FOM), which is part of the 'Netherlands Organisation for Scientific Research' (NWO). B.N.M. is grateful for a Royal Society Wolfson Research Merit Award.

## Author contributions

B.N.M. initiated the work; S.C., N.S., K.S., B.R., G.M. and M.N. performed the experiments; S.C. and N.S. analysed the data; B.N.M. and P.T.G. provided theoretical tools; S.C., C.R.P., G.A. and B.N.M. wrote the paper.

## Additional information

**Competing interests:** The authors declare no competing financial interests.

**Publisher's note**: 

