## [Peer Review File · Nature Communications]

Reviewers' comments:

Reviewer #1 (Remarks to the Author):

I have read "Coherent superpositions of three states for phosphorous donors in silicon prepared using THz radiation" and think this is an excellent interesting paper that should be published in Nature Comm.

These authors demonstrate THz laser pulse control of the relative phase and amplitude of multiple orbitals in Si:P and observe beat patterns produced by Zeeman splitting, which they resolve to the high fidelity of up to 95%.

My only negative comment is in the authors discussion of the formalism. They should be more clear about the definitions of t , t_p and t_d . In this regard, I think that there are some mistakes in the manuscript and SI, where these times are substituted for each other. In the SI, for instance, the equation of Ψ after the line "In the dark after the pulse.." I believe has some mistake where t_d should be t . The authors should check all these symbols for inconsistencies.

Otherwise I think this is an excellent paper. For the first time control of multiple impurity orbitals has been demonstrated. The physics is clear and (except for the issue discussed above) presented excellently.

Reviewer #2 (Remarks to the Author):

Review of Chick et al.

Chick et al. demonstrate the control of orbital states of donors in silicon. In particular, they show that they are able to achieve significant control of a three state superposition consisting of the $1s$ and $2p_-$, $2p_+$ states. Such control is potentially very important, allowing as it would directional selection of exchange coupling between a pattern of donors. This could, as the authors suggest, form the basis of a possible surface code architecture for quantum computing.

The experimental work is rather challenging, since the orbital transitions of phosphorus in silicon lie in the THz range, and hence the work must be carried out using a Free Electron Laser source (this might be an issue for future quantum computing, but I'm happy to leave that as a future challenge beyond the scope of this paper...).

I feel this is significant and technically demanding work, which I enjoyed reading, and I would like to move it towards publication in Nature Communications. I do however, have some questions and suggestions which I would like to see addressed before I can make a final recommendation.

My most significant concern is that the authors show data produced using x-polarized pulses alone - and interpret their data as arising from interference between the two circularly polarised field split eigenstates. If this interpretation is correct, then presumably a circularly polarised pulse pair would generate no such interference, and would also allow for more control over the resulting orbital wavefunction. (The authors do mention that changing polarisation would allow further control, but don't demonstrate it). Could the authors provide more data, with different polarisations, to back up their interpretation, and provide further control? I would also like to see electrical current data taken at different fields, showing that the quantum beating of Fig 3 does indeed have its frequency changed linearly with field (and a plot of extracted beat frequency against field).

Note that the authors have already published work showing the coherent generation of superpositions of both $1s$ with $2p_0$, and ($2p_-$ and $2p_+$). The new aspect of the work they do here is the magnetic field work, and so I do think this needs a more thorough investigation before publication can be recommended.

Other, more routine suggestions are as follows:

1. Line 92: "most of the hyper spherical surface": what does "most of" mean here?
2. Line 103: can the authors be more explicit that the Fourier transforms are with respect to the time delay t_d . Also, do they average their time scans over the time t ?
3. Fig 2. I note here that field dependence of transition energy is shown - but why it is necessary to include the $3p_0$ transition here? why not zoom in more closely on the $2p$ lines?
4. Line 109 - free electron laser (typo).
5. Lines 123-124: can the authors clarify their statements about background here?
6. Fig. 1: state time units (presumably ps).
7. Fig. 1: label x,y,z axes on the figures, and include at least one scale in nm.
8. Fig. 2. include x -axis label on right hand figures.

The authors present experimental measurements of the response of phosphorus donors in Silicon, in a relatively small magnetic field, to a pair of pulses of narrow-band THz radiation created by the Dutch free-electron laser FELIX tuned to a frequency near the 1s-2p⁺ and 1s-2p⁻ levels of this donor. The magnetic field splits the degeneracy of the 2p⁺ and 2p⁻ levels, which are degenerate in zero field. As a result, the photoresponse exhibits a splitting in frequency space, and quantum beats in the time domain. The experimental results very clearly demonstrate the splitting and the quantum beats, and hence clearly support the claim of the title—the creation of coherent superpositions of three states for phosphorus donors in silicon prepared using THz radiation.

The authors introduce their experimental results with a discussion of superposition of three orbital states, and present some calculations of the time-evolution of hydrogenic orbitals. The authors claim that their experimental results could lead to a simplification of existing surface code networks.

The manuscript is seriously deficient in its citations of earlier relevant work. The calculations presented in Fig. 1 are also too simplistic to be of more than purely illustrative value. They neglect well-understood semiconductor physics. Both of these deficiencies need to be addressed in the manuscript before it is published.

1. Citations:

- a. *Previous studies of donors in Silicon.* By design, and as the authors point out, the experiments are performed in the limit of weak excitation, so that the experimental observations (the frequency of and splitting between the levels of interest) could be predicted based on measurements using a standard spectroscopy technique, Fourier Transform Infrared Spectroscopy (FTIR). The manuscript neglects to reference a very rich history of the study of defects in semiconductors using FTIR in general, and photothermal ionization spectroscopy in particular. Particularly important are the initial report of the Zeeman effect of impurity levels in Silicon by Zwerdling, Button and Lax (Phys. Rev. 118, 975 (1960)), which were made for the Bi impurity, and a later FTIR study of Phosphorus donors in a magnetic field (Phys. Rev. B 48, 10864 (1993), and refs. 7 and 8 therein).
- b. *Previous coherent THz studies of Hydrogenic systems in semiconductors.* The manuscript is also missing some important citations to other coherent THz measurements on Hydrogenic systems in semiconductors. Particularly relevant are: *Coherent Manipulation of Semiconductor Quantum Bits with Terahertz Radiation*, by B. E. Cole, J. B. Williams, B. T. King, M. S. Sherwin, and C. R. Stanley (Nature 410, 60 (2001)), in which Rabi oscillations were reported on the 1s-2p⁺ transition of Hydrogenic donors in GaAs in a magnetic field, and *Quantum Coherence in an Optical Modulator*, by S. G. Carter, V. Birkedal, C. S. Wang, L. A. Coldren, A. V. Maslov, D. S. Citrin, and M. S. Sherwin (Science 310, 651 (2005)), the 1st observation of the excitonic Autler-Townes effect.
- c. The paper in general, and the calculations in particular, also neglect the very important and interesting complication that, since Silicon has 6 degenerate valleys, the Hydrogenic theory must be modified, as was originally done by Kohn

and Luttinger in 1955. This is an important distinction to the case of shallow impurities in GaAs, which has only a single valley and for which a Hydrogenic model, with only the effective mass and dielectric constant altered from their vacuum values, reproduce experimental observations extremely accurately, with negligible central cell correction. The Kohn-Luttinger theory also explains why the different species of impurities have different ground state energy levels, which is important to the implementation of surface code as the authors propose.

2. *Semiconductor physics*: The results of calculations presented in Fig. 1 are too simplistic to be useful—they don't contribute more than could be understood from a purely conceptual diagram. It would be much more interesting to see calculations in which the wave functions were derived from theory of Kohn and Luttinger (1955) for donors in Silicon. Such calculations would significantly strengthen the case for the potential of orbital control of donors in Silicon to contribute to quantum computing in the way that the authors claim. In addition, in these figures, the neighboring impurities are placed ± 2 atomic units away from the central impurity, where atomic units are, I assume, renormalized Bohr radii. This is unphysical—these impurities will be substitutional, registered with the Silicon lattice. The lattice constant of Silicon needs to be represented somehow.
3. A minor point: why was the magnetic field oriented in the 111 direction?

In summary, the manuscript presents interesting measurements that demonstrate, for what I believe is the first time, coherent superposition of three states for phosphorus donors in Silicon prepared using THz radiation. The manuscript also makes the interesting claim that the ability to coherently manipulate orbital states of this impurity with THz radiation may be useful for Silicon-based quantum computing. However, the manuscript neglects citations to important previous work, and the simulations presented are not sufficiently realistic to be compelling. Both of these deficiencies must be corrected before this article is accepted for publication.

Mark Sherwin.

Dear Sir/Madam,

We thank the referees for their careful reading of our manuscript and their helpful suggestions. We have acted on them, and we think this has resulted in a greatly improved paper.

We list the referees comments and detailed responses below. We hope you find it acceptable,

Best regards,
Ben

Reviewers' comments:

Reviewer #1 (Remarks to the Author):

I have read "Coherent superpositions of three states for phosphorous donors in silicon prepared using THz radiation" and think this is an excellent interesting paper that should be published in Nature Comm.

These authors demonstrate THz laser pulse control of the relative phase and amplitude of multiple orbitals in Si:P and observe beat patterns produced by Zeeman splitting, which they resolve to the high fidelity of up to 95%.

My only negative comment is in the authors discussion of the formalism. They should be more clear about the definitions of t , t_p and t_d . In this regard, I think that there are some mistakes in the manuscript and SI, where these times are substituted for each other. In the SI, for instance, the equation of Ψ after the line "In the dark after the pulse.." I believe has some mistake where t_d should be t . The authors should check all these symbols for inconsistencies.

We accept that this was inconsistent and have corrected that formula and the definitions of symbols t and t_d , new version, Eqn 1&2.

Otherwise I think this is a excellent paper. For the first time control of multiple impurity orbitals has been demonstrated. The physics is clear and (except for the issue discussed above) presented excellently.

Reviewer #2 (Remarks to the Author):

Review of Chick et al.

Chick et al. demonstrate the control of orbital states of donors in silicon. In particular, they show that they are able to achieve significant control of a three state superposition consisting of the $1s$ and $2p^-$, $2p^+$ states. Such control is potentially very important,

allowing as it would directional selection of exchange coupling between a pattern of donors. This could, as the authors suggest, form the basis of a possible surface code architecture for quantum computing.

The experimental work is rather challenging, since the orbital transitions of phosphorus in silicon lie in the THz range, and hence the work must be carried out using a Free Electron Laser source (this might be an issue for future quantum computing, but I'm happy to leave that as a future challenge beyond the scope of this paper...).

We agree with the referee that a technology that relies on an FEL for operation is somewhat impractical. Having now made considerable effort in the face of the challenges of working in the THz to establish the characteristics of the pulses required for coherent control, we are now in the process of developing a table-top system. We agree with the referee that this is beyond the scope of the present work.

I feel this is significant and technically demanding work, which I enjoyed reading, and I would like to move it towards publication in Nature Communications. I do however, have some questions and suggestions which I would like to see addressed before I can make a final recommendation.

My most significant concern is that the authors show data produced using x-polarized pulses alone - and interpret their data as arising from interference between the two circularly polarised field split eigenstates. If this interpretation is correct, then presumably a circularly polarised pulse pair would generate no such interference, and would also allow for more control over the resulting orbital wavefunction. (The authors do mention that changing polarisation would allow further control, but don't demonstrate it). Could the authors provide more data, with different polarisations, to back up their interpretation, and provide further control?

The addition of variable polarization makes the Ω -terms in the time-dependence matrix different, giving control over the relative phase and relative magnitude, and theoretically allows complete control over the 3D Hilbert space (within the approximation that $\Delta \ll \omega$ mentioned on L92). With only a pair of x-polarised pulses, nearly the complete Hilbert space spanned by the 3D state-vector of unit norm can be covered; only very small regions near $P_{2p_{\pm}}=1$ are inaccessible (see response to the referee's point 1 below for more details). We previously demonstrated that polarization controls the disappearance of one of the $2p_{+}/2p_{-}$ doublet depending on the sign of the field (fig 2c of ref [16]) producing a 2D Hilbert space, and we already demonstrate here (Fig 6-Middle) that the beats are removed when moving from 3D to 2D Hilbert space. The aim of this work was to demonstrate superpositions in three-level systems via the beats, and since removing the beats is easy, demonstrating their removal in a sophisticated way seemed unnecessary.

I would also like to see electrical current data taken at different fields, showing that the quantum beating of Fig 3 does indeed have its frequency changed linearly with field (and a plot of extracted beat frequency against field).

Agreed - we show the corresponding electrical experimental data for the same fields and added a new Fig 7 and surrounding text. The data are at lower resolution and lower statistical power than Fig 6 (as mentioned in the caption), and so the error bars on the frequency of each peak is higher.

Note that the authors have already published work showing the coherent generation of superpositions of both 1s with 2p₀, and (2p₋ and 2p₊). The new aspect of the work they do here is the magnetic field work, and so I do think this needs a more thorough investigation before publication can be recommended.

We agree with the referee that the magnetic field is the new aspect, and the reason this aspect is important is because it produces a pair of excited states with dipole allowed transitions within the laser pulse bandwidth. As mentioned above, we already demonstrated that changing the polarization simply reduces the situation back to a single excited state, and that a 2D Hilbert space does not produce beats. We have now added an entirely new section on donor wavefunctions in Si in a small (our case) magnetic field.

Other, more routine suggestions are as follows:

1. Line 92: "most of the hyper spherical surface": what does "most of" mean here?
Agreed, this was vague and we have now state L95-96 "The value of $|a_0|^2$ can be varied from 0 to 1, but the maximum $|a_{\pm}|^2$ is 27/32."
2. Line 103: can the authors be more explicit that the Fourier transforms are with respect to the time delay t_d . Also, do they average their time scans over the time t ?
Agreed, on L176 we add the FT was "(with respect to t_d)". In Eqn 2, a time scan depending on t would require a third projection pulse.
3. Fig 2. I note here that field dependence of transition energy is shown - but why it is necessary to include the 3p₀ transition here? why not zoom in more closely on the 2p lines?
It was not included for any other reason than to keep the abscissa scale the same as the main panel. We have made the zoom as suggested (see new Fig 4 inset) and included error bars to indicate the uncertainty due to datapoint spacing.
4. Line 109 - free electron laser (typo).
Agreed L183
5. Lines 123-124: can the authors clarify their statements about background here?
We expanded this remark L202-205: "This is for the simple reason that the product of the transmission spectrum with the system response spectrum (as in Fig 4) contains a complex background, while the product of the absorption spectrum and the system response spectrum is background free, i.e. it contains just the atomic transition features"
6. Fig. 1: state time units (presumably ps).
Agreed. The time units were in the caption, but (following the advice of the third referee) the Si:P temporal response, and these times, are now on a new figure 3 "The labelled times t_d are shown in ps"
7. Fig. 1: label x,y,z axes on the figures, and include at least one scale in nm.
We have included scale bars on each part of Fig 1 a-d. (and new Fig 3)

8. Fig. 2. include x-axis label on right hand figures.

We have included labels on the scale bars in each part of Fig 1 a-d. (and the new Fig 3)

Reviewer #3

The authors present experimental measurements of the response of phosphorus donors in Silicon, in a relatively small magnetic field, to a pair of pulses of narrow-band THz radiation created by the Dutch free-electron laser FELIX tuned to a frequency near the $1s:2p+$ and $1s:2p-$ levels of this donor. The magnetic field splits the degeneracy of the $2p+$ and $2p-$ levels, which are degenerate in zero field. As a result, the photoresponse exhibits a splitting in frequency space, and quantum beats in the time domain. The experimental results very clearly demonstrate the splitting and the quantum beats, and hence clearly support the claim of the title—the creation of coherent superpositions of three states for phosphorus donors in silicon prepared using THz radiation.

The authors introduce their experimental results with a discussion of superposition of three orbital states, and present some calculations of the time-evolution of hydrogenic orbitals. The authors claim that their experimental results could lead to a simplification of existing surface code networks. The manuscript is seriously deficient in its citations of earlier relevant work. The calculations presented in Fig. 1 are also too simplistic to be of more than purely illustrative value. They neglect well understood semiconductor physics. Both of these deficiencies need to be addressed in the manuscript before it is published.

We accept these criticisms and have improved the manuscript in response to each of the detailed comments below.

1. Citations:

a. *Previous studies of donors in Silicon.* By design, and as the authors point out, the experiments are performed in the limit of weak excitation, so that the experimental observations (the frequency of and splitting between the levels of interest) could be predicted based on measurements using a standard spectroscopy technique, Fourier Transform Infrared Spectroscopy (FTIR). The manuscript neglects to reference a very rich history of the study of defects in semiconductors using FTIR in general, and photothermal ionization spectroscopy in particular. Particularly important are the initial report of the Zeeman effect of impurity levels in Silicon by Zwerdling, Button and Lax (Phys. Rev. 118, 975 (1960)), which were made for the Bi impurity, and a later FTIR study of Phosphorus donors in a magnetic field (Phys. Rev. B 48, 10864 (1993), and refs. 7 and 8 therein).

Agreed. We have included these references in our discussion on LL36-38.

b. *Previous coherent THz studies of Hydrogenic systems in semiconductors.* The manuscript is also missing some important citations to other coherent THz measurements on Hydrogenic systems in semiconductors. Particularly relevant are: *Coherent Manipulation of Semiconductor Quantum Bits with Terahertz Radiation*, by B. E. Cole, J. B. Williams, B. T. King, M. S. Sherwin, and C. R. Stanley (Nature 410, 60 (2001)), in which Rabi oscillations were reported on the $1s:2p+$ transition of Hydrogenic donors in GaAs in a magnetic field, and *Quantum Coherence in an Optical Modulator*, by S. G. Carter, V. Birkedal, C. S. Wang, L. A. Coldren, A. V. Maslov, D. S. Citrin, and M. S. Sherwin

(Science 310, 651 (2005)), the 1st observation of the excitonic Autler-Townes effect.
Agreed. We have included these references in our discussion on LL44-47.

c. The paper in general, and the calculations in particular, also neglect the very important and interesting complication that, since Silicon has 6 degenerate valleys, the Hydrogenic theory must be modified, as was originally done by Kohn and Luttinger in 1955. This is an important distinction to the case of shallow impurities in GaAs, which has only a single valley and for which a Hydrogenic model, with only the effective mass and dielectric constant altered from their vacuum values, reproduce experimental observations extremely accurately, with negligible central cell correction. The Kohn-Luttinger theory also explains why the different species of impurities have different ground state energy levels, which is important to the implementation of surface code as the authors propose.

Agreed. This is as the referee says, complicated, and we are pleased that the referee's advice is to include it for the general readership of Nature Communications. We have therefore added major new sections LL98-162, LL437-498, and new Figs 2-3 to expand on the similarities and differences between hydrogen and silicon donors. The section includes original work such as the theory for the donor selection rules with dipole moment matrix element values, and simulations for the multi-valley superposition evolution resulting from a general excitation pulse.

2. *Semiconductor physics:* The results of calculations presented in Fig. 1 are too simplistic to be useful—they don't contribute more than could be understood from a purely conceptual diagram. It would be much more interesting to see calculations in which the wave functions were derived from theory of Kohn and Luttinger (1955) for donors in Silicon. Such calculations would significantly strengthen the case for the potential of orbital control of donors in Silicon to contribute to quantum computing in the way that the authors claim. In addition, in these figures, the neighboring impurities are placed ± 2 atomic units away from the central impurity, where atomic units are, I assume, renormalized Bohr radii. This is unphysical—these impurities will be substitutional, registered with the Silicon lattice. The lattice constant of Silicon needs to be represented somehow.

Agreed. We have added new figs 2&3 with full donor wavefunctions. Regarding the ± 2 atomic units, this is simply a matter of rounding. The renormalized Bohr radius is 3nm (L36, L452), and the lattice period (0.56nm) is therefore 0.18 Bohr. We chose to indicate the sites of neighbours at 10 lattice periods = 1.8 Bohr which we rounded to 2 (1.8 and 2 units are indistinguishable on the figure).

3. A minor point: why was the magnetic field oriented in the 111 direction?

We now explain this on LL138-148: "In the case of a magnetic field..... other field directions produce a more complex spectrum".

In summary, the manuscript presents interesting measurements that demonstrate, for what I believe is the first time, coherent superposition of three states for phosphorus donors in Silicon prepared using THz radiation. The manuscript also makes the interesting claim that the ability to coherently manipulate orbital states of this impurity with THz radiation may be useful for Silicon-based quantum computing. However, the

manuscript neglects citations to important previous work, and the simulations presented are not sufficiently realistic to be compelling. Both of these deficiencies must be corrected before this article is accepted for publication.

REVIEWERS' COMMENTS:

Reviewer #2 (Remarks to the Author):

I have read the response of the authors to my first report and I am pleased to see improvements in several respects. The only outstanding issue is that the authors still do not show the multi-pulse control with different polarization, which I do continue to believe would make a powerful addition to the story. It would allow the entire Hilbert space to be accessed using a single protocol type.

However, given the impressive achievements already demonstrated in this paper, in what is a technically demanding experiment, and given that the authors do indeed show polarization control, albeit with a simpler method in a previous paper, I have on balance decided that the paper can be published without further delay.

Reviewer #4 (Remarks to the Author):

I really enjoyed reading the revised paper. In response to my referee report, the authors have done a significant amount of theoretical work to calculate the time-evolution of the wave functions of Si donors using the Kohn-Luttinger theory. In my opinion, these results add a lot to the paper. I also really like the description of how the results of this paper might lead to the implementation of a surface code in a Si-based quantum computer.

My only suggestion is the following: in Fig. 3c, the color scale of the heat map makes it extremely difficult to spot the differences between between wave functions plotted in a given row. I would recommend changing the color scale to make the differences more clear (perhaps a logarithmic color scale? I leave it to the authors).

I presume that the reason the contrast is reduced in Fig. 3c compared with Fig. 1 is the smaller value of α . I would recommend adding a clause to that effect near lines 153-156.

I recommend publication in Nature Communications with these minor optional revisions.

Response to referees

REVIEWERS' COMMENTS:

Reviewer #2 (Remarks to the Author):

- I have read the response of the authors to my first report and I am pleased to see improvements in several respects. The only outstanding issue is that the authors still do not show the multi-pulse control with different polarization, which I do continue to believe would make a powerful addition to the story. It would allow the entire Hilbert space to be accessed using a single protocol type.
- However, given the impressive achievements already demonstrated in this paper, in what is a technically demanding experiment, and given that the authors do indeed show polarization control, albeit with a simpler method in a previous paper, I have on balance decided that the paper can be published without further delay.

We acknowledge that the reviewer is enthusiastic for the story to be complete, and are grateful that they consider the manuscript to be suitable in its current form.

Reviewer #4 (Remarks to the Author):

- My only suggestion is the following: in Fig. 3c, the color scale of the heat map makes it extremely difficult to spot the differences between between wave functions plotted in a given row. I would recommend changing the color scale to make the differences more clear (perhaps a logarithmic color scale? I leave it to the authors).

We have carefully considered this suggestion, and concluded that a linear colourbar is a more intuitive way of communicating the wavefunction's shape. The scale of the changes are indeed smaller in Fig 3(c) than Fig 1, and we believe that this is an important aspect to communicate visually. We have extended the discussion a little to explain this (see next comment).

We also acknowledge that part of the reason these variations were hard to spot was due to an error on the part of the authors: one of the tiles was omitted, and a copy of the previous timestep inserted in its place. This error has been corrected and the other figures checked again, and we are grateful to the reviewers for drawing our attention back to the figure.

- I presume that the reason the contrast is reduced in Fig. 3c compared with Fig. 1 is the smaller value of alpha. I would recommend adding a clause to that effect near lines 153-156.

The difference in contrast is due to the central cell correction, which makes the ground state smaller compared to the excited state, and thus reduces the obviousness of this effect in the picture. We have added a sentence to that effect on lines 179-180.